

# Automated urban landscape design: an AI-driven model for emotion-based layout generation and appraisal

Xiaohu Tang and Won-jun Chung

Department of Visual Communication Design, Tongmyong University, Busan, Republic of South Korea

## ABSTRACT

The evolution of a city is significantly shaped by the design of its urban landscape. The advancement of artificial intelligence has substantially increased convenience for individuals. This research proposes an urban landscape layout model powered by artificial intelligence that automatically generates urban landscape design based on deep learning (URDDL) with two dimensions: emotional tendency and urban landscape appraisal. The input image represents land use and surrounding road conditions, while the output image depicts the selection of the main entrance and the internal spatial function layout. The Pix2Pix model is trained to learn the internal function layout based on varying land use and road conditions. Additionally, a domain-specific dictionary is constructed using an existing semantic resource vocabulary, where positive and negative sentiment words are compared with their corresponding sentiment values, focusing on categories such as Stimulate, Sense, and Action. Experimental results indicate that the absolute average error of the URDDL model is 91.31%, with a maximum error of 96.87%. The degree of fit is highly appropriate for evaluating the emotional prediction of urban landscapes. The findings demonstrate that the URDDL model outperforms traditional design methods regarding generated results, suggesting its potential for future applications in automated landscape design.

# INTRODUCTION

In urban settings, residents frequently engage with urban landscapes as natural spaces, which serve as venues for rest, recreation, exercise, and social interaction, fulfilling their leisure needs while providing essential social services (*Marvuglia et al., 2020*; *Natanian & Auer, 2020*). A well-designed urban landscape is a key component of green infrastructure, embodying wellness, suitability, fairness, and versatility. The concept of scale plays a crucial role in urban landscape design, influencing everything from spatial layout to the arrangement of physical elements and the orderly functioning of these spaces. However, in recent years, there has been a noticeable trend toward homogenization in urban landscape construction (*Wu et al., 2021*; *Rybarczyk et al., 2020*), and this often results in designs that are overly planar and need to adequately address the nuanced requirements for scale that users might desire. Contemporary urban green space design emphasizes a people-centered

Corresponding author
Won-jun Chung,
wonjchung@163.com

approach, aiming to create an inviting urban environment that aligns with human needs. The design of the urban landscape layout is critical in the preliminary stages of planning, guiding the direction of more detailed design work.

A rational urban landscape layout must consider not only the positioning of key entrances and the functional arrangement of buildings such as educational facilities, sports fields, and gymnasiums but also the relationships between these elements. Furthermore, factors such as noise distance, height restrictions, and the orientation of buildings must be meticulously evaluated, often relying on the extensive experience of architects who use iterative calculations and comparisons to arrive at an optimal layout that satisfies multiple requirements. With the advent of artificial intelligence technology, particularly the advanced image-processing capabilities of deep learning, the potential for machines to learn from existing case studies and autonomously generate design solutions has become a significant focus of our research (*Abo-El-Wafa, Yeshitela & Pauleit, 2018*; *Taoufiq, Nagy & Benedek, 2020*).

With interdisciplinary technical support, we try to explore new ways of machine-aided architectural design. At the end of the 20th century, it has made a lot of research on the ecological landscape of East Asia and North America. In terms of theoretical research, since 2000, pragmatism philosophy, intuitionistic phenomenology, cybernetics, *etc.*, have been used as the basis of ecological aesthetics to demonstrate the relationship between ecology and aesthetics, including the specific way of integrating ecological aesthetics into landscape environment experience, the application of ecological aesthetics in the management process, and the expression of ecological ideas in art design. In 2019, *Qin et al. (2019)* put forward ecological aesthetics and its perception mode by combining the progress of the Western aesthetic system based on the ideological origin of traditional aesthetics with relevant concepts of landscape design, which laid a foundation for the connotation research of ecological aesthetics in landscape. *Law et al. (2020)* laid a foundation for the involvement of ecological aesthetics in the field of landscape design. *Zhao et al. (2018)*, combined with the theories and thoughts of Western scholars, further explained the basic categories of ecological aesthetics and made valuable contributions to the discipline construction. To sum up, the problem of disunity of wetland ecosystems in urban wetland landscapes has been explored in many aspects in past research, but there is still a lack of communication mechanism between the relevant research of landscape perception and the landscape design research under the guidance of ecological, aesthetic values, and the potential contradiction between them has not been solved completely.

Urban landscape design plays a crucial role in shaping the evolution of cities, yet traditional design methods often need to be revised to address the complexities of modern urban environments. Despite the significant advancements in artificial intelligence (AI), there remains a gap in the integration of AI-driven models explicitly tailored for urban landscape design that accounts for both functional and emotional dimensions. The purpose of this research is to bridge this gap by proposing an innovative Urban Landscape Layout Model (URDDL) powered by deep learning, which automatically generates urban landscape designs. This model introduces a dual-dimensional approach, incorporating emotional tendencies and urban landscape appraisal into the design process.

The main contributions of this study include the development of a model that inputs land use and road conditions to output spatial function layouts and main entrance selections using the Pix2Pix model trained on varying scenarios. Furthermore, a domain-specific sentiment dictionary enhances the model's capacity to evaluate emotional responses to urban landscapes, focusing on key categories such as Stimulate, Sense, and Action. This study not only advances the field of automated landscape design but also sets the stage for future AI applications in urban planning, offering a more nuanced and efficient approach to creating emotionally resonant urban environments.

## RELATED WORKS

With the rapid development of technology and the auxiliary role of machines in the design process, artificial intelligence, automation, and machine learning are becoming inevitable opportunities and challenges in almost all disciplines, including architecture. In the research before *Zhao et al. (2018)*, pix2pix was used to expand the scope of flat plan generation. By learning from Boston's single residence, the complete process of building plane design generation was realized from land use to building exterior contour, from external contour to building internal functional color block layout, and finally to furniture generation. At the same time, it discusses the possibility of extending the scheme to urban design, which has a high reference value. In recent years, the research and application of deep learning have grown explosively. It has made significant achievements in face recognition (*Kim, Jain & Liu, 2022*), speech and image processing (*Bakhshi, Harimi & Chalup, 2022*), language translation (*Camgoz et al., 2020*; *Lin et al., 2024*), automatic driving (*Kotwal, Kashyap & Shafi, 2024*), and other aspects. It has made some research achievements in the elements of architectural style migration, apartment plane recognition and generation, building facade generation, and so on. *Jeong et al. (2022)* used the scale method mainly to quantify the specific items in the planning and design and then get the corresponding scores from the users. Such a survey method can fully cover the content of planning and design, and the survey results are relatively intuitive. The image analysis method shows the interviewees different types of images. It investigates the impact and feelings of the environment on people from the aspects of environmental psychology so as to understand the evaluation of different environments.

These methods ignore the subjectivity of people's perception process and the consumption process and overestimate people's willingness to pay. *He et al. (2021)* and *Pan et al. (2024)* put forward that different socio-economic groups have significant differences in willingness to pay or have the same willingness to pay, but the ability to pay is different. Moreover, it is inherently risky to assign monetary value to public goods (such as ecosystem services) that are prone to change with their followers. In 2018, *Espinosa, Velastin & Branch (2018)* applied a Bayesian network to learn a large number of flat bubble diagram samples of residential buildings. By counting the conditional probability of each factor relationship, the automatic generation of a bubble diagram of the residential layout was realized. At the same time, other algorithms were applied to finally realize the one-click generation from functional bubble diagram to 2D plan and then to 3D model. Since Bayesian networks
are designed with less or less artificial input, it is often necessary to explore a large amount of artificial knowledge at the core of Bayesian networks. In 2019, *Karimzadeh et al. (2021)* proposed the generation countermeasure neural networks (GANS), which is a deep learning model using neural networks to generate images. Their design can be seen as the design of images and ultimately building entities in reality based on images. The proposal of GANS also makes the construction industry try to apply artificial intelligence to architectural design from the perspective of image. In 2018, *Dominguez-Sanchez, Cazorla & Orts-Escolano (2018)* continued to study from the image direct learning proposed pix2pix based on the conditional generation countermeasure network (CGAN). They realized the generation of street view and building elevation based on the marked image in the article. *Lin et al. (2021)* built a more detailed network, pix2pixhd, on the basis of pix2pix and improved the image resolution from 256 * 256 to 2048 * 1024. The results show better street view generation and interior design generation based on tagged images.

Since AI has increasingly influenced urban planning, offering transformative solutions to traditional design challenges. *Jiang et al. (2024)* review various AI techniques applied to urban planning, including generative models and predictive analytics. They highlight the use of GANs for automating design tasks and optimizing spatial layouts. Despite their promising capabilities, they note that many AI methods, including those reviewed, often need more integration with the emotional and functional dimensions of urban design. This gap suggests an opportunity for further research, particularly in models that address both aesthetic and functional aspects simultaneously.

*Senem et al. (2023)* specifically examine deep learning models used in urban landscape design, with a focus on the Pix2Pix model. Pix2Pix is a conditional GAN model known for its ability to generate high-quality images from input data, such as land use and road conditions. Their study demonstrates Pix2Pix's efficacy in producing realistic urban layouts but also acknowledges limitations, such as sensitivity to variations in input data and the challenge of incorporating complex emotional responses into the design process. This critique underscores the need for models that not only generate functional designs but also consider the emotional impact on users.

*Wang (2023)* provide insights into the emotional impact of urban environments and methods for incorporating emotional responses into design. Their research utilizes sentiment analysis techniques to evaluate how urban spaces affect residents' emotions. They critique existing methods for their limited scope in capturing nuanced emotional experiences and suggest integrating more comprehensive sentiment analysis into design models. This perspective is relevant for enhancing models like URDDL, which aims to combine emotional and functional design dimensions.

*Bordoloi & Biswas (2023)* explore the use of semantic resources and domain-specific dictionaries in design models. Their study focuses on how sentiment dictionaries can improve model accuracy by providing contextually relevant sentiment values. They critique the simplistic application of semantic resources in some design models and advocate for more sophisticated approaches that consider contextual variations in sentiment analysis. This critique is pertinent to the URDDL model's approach, which utilizes a domain-specific dictionary to assess emotional tendencies in urban landscapes.

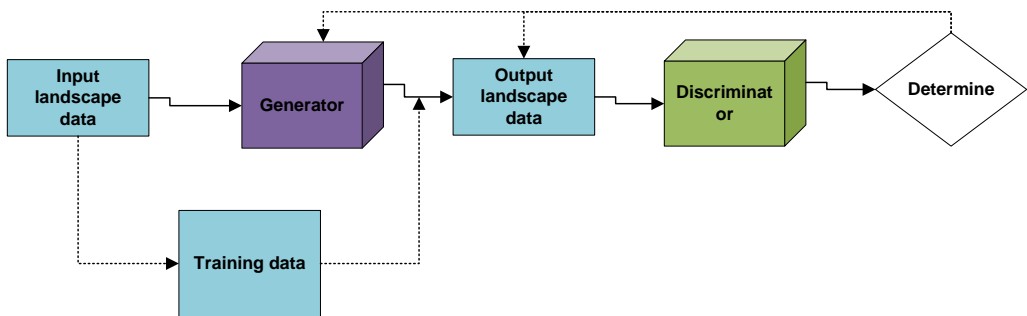

**Figure 1** The modeling process.

*Şekerci et al. (2024)* and *Song et al. (2024)* discuss recent innovations in AI-driven urban landscape design, including automated design generation and emotion-sensitive models. They highlight the potential of integrating AI technologies to enhance design processes but also note that many models, including those reviewed, still face challenges in fully capturing the emotional and functional complexity of urban environments. Their critique emphasizes the need for continuous improvement in AI models to address these challenges comprehensively.

These studies collectively provide a detailed examination of AI's role in urban landscape design, focusing on specific models such as Pix2Pix and the integration of emotional analysis.

# URDDL MODEL

## Basic modeling

In the URDDL scheme, the whole experimental study is mainly divided into three processes, as shown in Fig. 1:

 (i) Training pix2pix model. Firstly, the selected samples are divided into a training set and a test set by manual annotation, both of which are paired images. The input image of the training set is the landscape land use and surrounding road conditions. The output image is the landscape primary entrance selection and internal spatial function layout. The pix2pix model is trained for the internal function layout ability of different land use and road conditions.

(ii) The trained generator is used to generate false samples. At the same time, the paired images in the test set are input to the generator g. The generator automatically generates the matching landscape interior layout results according to the training rules. It juxtaposes the results with the original paired images to intuitively compare the differences between the generated results and the actual results.

(iii) Qualitative and quantitative analysis of the generated false samples and actual samples. From the point of view of a conventional building plan investigation, the created products are compared with the genuine products, and the quality of the machine produced comes about is assessed.

Data preparation is used to produce training data sets. The quality of training data sets is essential for deep learning training. Only by providing enough good data sets can the pix2pix model produce satisfactory results. The model training involves the input of the prepared data set into the established training environment, and the computer automatically completes the training process by adjusting a few parameters.

The generator $G$ in the URDDL scheme is designed to convert input images into generated outputs that approximate the desired results. It utilizes an encoder–decoder architecture, with the encoder progressively down-sampling the input image to extract high-level features. At the same time, the decoder up-samples these features to produce the output image. Specifically, the pix2pix model employs a U-Net architecture characterized by its inclusion of skip connections that preserve spatial information by linking corresponding encoder and decoder layers. The input to the generator is a paired image representing landscape land use and surrounding road conditions. At the same time, the output is a synthesized image depicting the internal spatial function layout and landscape entrance selection. The generator's training involves minimizing the discrepancy between generated images and real-world images from the training set. This process employs a combination of pixel-wise loss, such as mean squared error (MSE) or L1 loss, and an adversarial loss. The adversarial loss is provided by the discriminator, guiding the generator to produce images that are increasingly indistinguishable from real samples through iterative optimization using gradient descent techniques.

The discriminator $D$ serves the essential function of distinguishing between real images and those generated by the generator. It is structured as a deep convolutional neural network (DCNN) with successive convolutional layers that reduce spatial dimensions while enhancing feature map richness. The discriminator processes both real images derived from the test set and generated images to assess their authenticity. The output of the discriminator is a probability score that indicates the likelihood of the input image being real. This score plays a crucial role in training the generator, as it provides feedback on the realism of the generated images. The discriminator's loss function is usually binary cross-entropy, which measures the accuracy of the discriminator in classifying images as real or fake. During training, the discriminator and generator are optimized concurrently. While the generator strives to produce more realistic images, the discriminator improves its capacity to distinguish between real and generated images, driving both components toward an equilibrium where the generator's outputs are nearly indistinguishable from genuine images.

The final result evaluation is the qualitative and quantitative analysis of results. The production of data sets also includes three steps:

- Data collection. The final purpose and effect of our experiment should be considered. The data collected in the first step should be filtered, and the samples that meet the screening principle should be retained as the basis of the third step of data annotation processing.
- Data processing to ensure the unity of training data style, on the basis of the second step of data filtering, the data is further labeled, and the labeled data is used as the final training data set of deep learning. Formula (1) is the final loss function, and Formula (2)

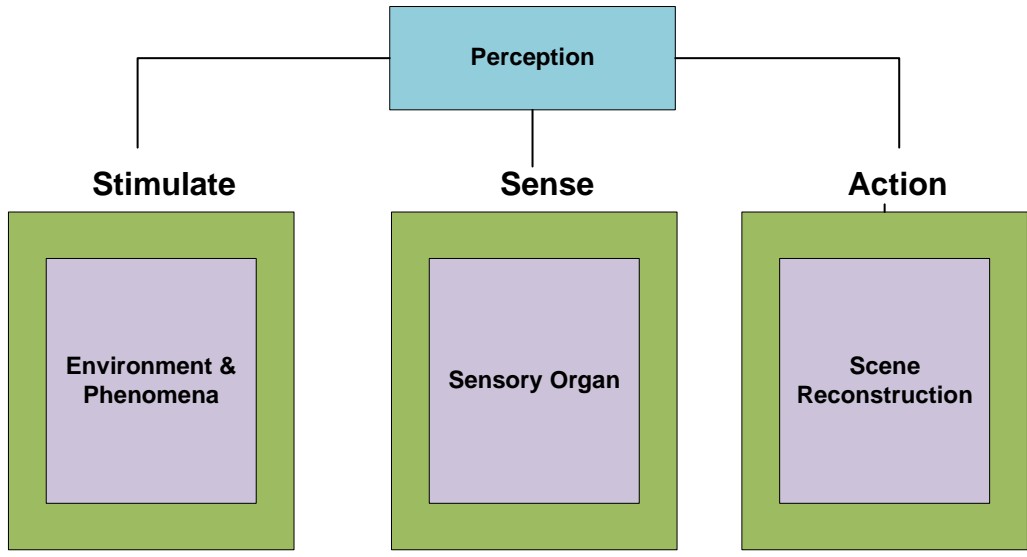

**Figure 2** Emotional perception.

is the position code for evaluating word order.

$$\sum y = nb_0 + b_1 \sum x_1 + b_2 \sum x_2, \quad \sum x_1 y = b_0 \sum x_1 + b_1 \sum x_1^2 + b_2 \sum x_1 x_2,$$
$$\sum x_2 y = b_0 \sum x_2 + b_1 \sum x_1 x_2 + + b_2 \sum x_2^2 \tag{1}$$

$$PE_{pos,2i} = \sin\left(\frac{pos}{10002i}\right)$$
$$PE_{pos,2i+1} = \cos\left(\frac{pos}{10002i}\right). \tag{2}$$

## Emotion perception

Emotional perception analysis can be regarded as a simple modeling of human emotional understanding (*Dzedzickis, Kaklauskas & Bucinskas, 2020*; *Gu et al., 2024*). In the evaluation of urban landscape design, a domain dictionary is created based on the existing semantic resource vocabulary, and the positive and negative emotional words are compared with corresponding emotional values, which are stimulating, sense, and action, as shown in Fig. 2.

URDDL mainly focuses on the interaction and influence mechanisms among space users, place, and emotion. The reason is that with the acceleration of urbanization and the improvement of life quality, the attention to spatial emotion helps to enhance residents' happiness and build a people-oriented smart city. The popularity of social media networks provides a channel for the expression of individual opinions. The mechanism of emotional spatial pattern is of great significance. By solving the above key problems, it can make suggestions and policy implications for decision-makers of urban public safety, public health, and design management.

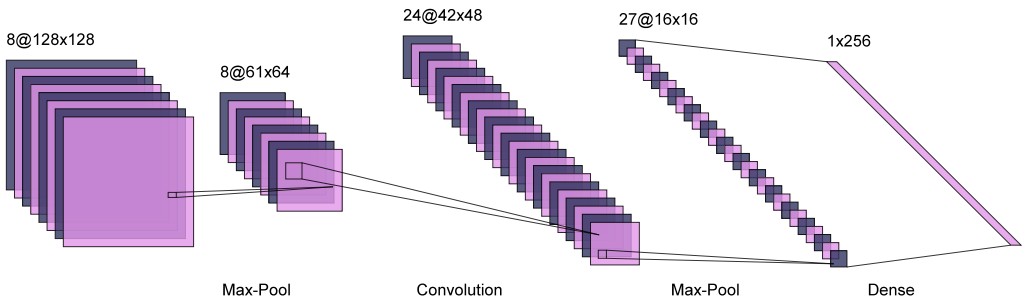

8@128x128
8@61x64
24@42x48
27@16x16
1x256

Max-Pool    Convolution    Max-Pool    Dense

**Figure 3    URDDL network.**

Users' emotions exist and can derive expectations, optimism, trust, and other emotions. These emotions cannot be summarized, and even complex emotions generated by the fusion of basic emotions are challenging to identify and determine by traditional methods. Because of this situation, when we embed emotional intelligence for URDDL, we assume that $\beta$ is the feature weight and $X$ Evaluate the word frequency in the data for users. Y is the number of neurons in the output layer.

$$\beta^1 = (X^T X)^{-1} X^T y = (\sum x_i x_{ii}^T)^{-1} (\sum x_i y_i). \tag{3}$$

In addition, in each kind of sentiment tendency classifier, the user comment text data represented by a word vector is used as the input of the model. The forward and backward semantic information of microblog comment Chinese text data is obtained through pix2pix, and the deep-seated feature vector is mined $v$. The probability distribution is shown in Formula (4) (*Takada, Wang & Yamasaki, 2021*).

$$p_i = pix2pix(w_C v + b) \tag{4}$$

where $P$ indicates the probability of user comments belonging to emotion $i$.

## Network data processing

For the layout generation of professional architecture and landscape, it is essential that the characteristics of graphics and the corresponding relationship between each other, rather than the pursuit of similarity between real samples, which is a significant difference between the two. Through the final generation results, we can also find that most of the generated results are different from the real samples, and these differences are just the performance of the machine to master its basic laws through learning a large number of samples. The specific network diagram is shown in Fig. 3.

The 280 pairs of samples obtained from data processing cannot be directly inputted to the computer for learning, but they need further processing. The training image size of the pix2pix model is 256 pixels * 256 pixels, and the result of data processing is 300 pixels/inch high-resolution image (*Harris et al., 2022*). The input to the model is a pair of corresponding images rather than a single image, so the data set preprocessing is also divided into two steps: image compression and image combination. Different training

purposes or training objects will have different optimal learning rates, and it is not a constant learning rate but a constant learning rate.

The essence of the training process is to update parameters, which requires each parameter to have a corresponding initial value. The selection of the initial value has a significant influence on the convergence speed and quality of the model. The number of iterations is the calculation times of each training sample. An iteration will calculate all the samples in the training set once. For example, if there are 25 training samples in the training set, each iteration will calculate the 25 samples one by one. Each calculation will update the generator $G$ and sum discriminator $D$ parameters in the first experiment; the training learning rate of the pix2pix model was 0.0002, the number of iterations was 2,200, and the training time was 1.5 h. Finally, the computer professional perspective to evaluate the quality of the generated results of the generated countermeasure network, one is based on the size of the loss function, the other is to compare the similarity between the generated results and the real samples, the more similar and more accurate, the better the training effect.

## EXPERIMENTS AND ANALYSIS

### Dataset

Given a target landscape land boundary and surrounding road conditions, the computer can automatically learn the layout rules of the general layout plan from the data set of the real landscape master plan and then get the design results. We use UBGGset (https://zenodo.org/records/8352777), which serves as a large-volume sample dataset specifically tailored to support and foster URD research endeavors. The UBGGset consists of 14,627 sample images (without data augmentation), with dimensions of 256 pixels in length and width, covering an urban area of approximately 2,272 km2. The UBGGset was constructed with co-registered pairs of 3 m planet images and fine-annotated urban landscapes labeled on 1 m Google Earth image. Input images are resized to $256 \times 256$ pixels and normalized. Normalization techniques such as mean subtraction and standardization are employed. Mean subtraction involves removing the mean pixel value from each color channel (red, green, blue), which is typically calculated across the entire training dataset. Standardization is achieved by dividing the pixel values by the standard deviation for each channel, helping to normalize pixel values to a consistent range.

After a large number of studies, it has been found that most of the landscapes in China are of similar regularity. With the requirements of the code, most construction projects in China have unilateral layouts, and the layout design results caused by the influence of local climate conditions are very similar to those in the Lingnan area. In the second experiment, the regional restrictions were relaxed, and the source of sample collection was extended to the whole country. At the same time, due to the large span between East and West, North and South, and the diversity of terrain, the scope is finally limited to the eastern and southern coastal and central areas of China. The types of landscape nodes are shown in Table 1, where "-" denotes unable to sum.

**Table 1  Comparison of different data sets.**

| Paper | Model | Training | Testing | Total |
|---|---|---|---|---|
| Jeong (*Bakhshi, Harimi & Chalup, 2022*) | Pix2Pix | 135 | 20 | 155 |
| He (*Camgoz et al., 2020*) | Pix2PixHD | 351 | 40 | 391 |
| Lin (*Espinosa, Velastin & Branch, 2018*) | Pix2Pix | 235 | 25 | 260 |

**Table 2  Parameter settings.**

| Parameter | Value |
|---|---|
| Model architecture | Pix2Pix |
| Optimizer | Adam |
| Learning rate | 0.0002 |
| Batch size | 1 |
| Number of epochs | 100 |
| Loss function | L1 loss + adversarial loss |
| L1 loss weight | 100 |
| Adversarial loss weight | 1 |
| Dropout rate (generator) | 0.5 |
| Dropout rate (discriminator) | 0.5 |
| Input image size | 256 × 256 pixels |
| Output image size | 256 × 256 pixels |
| Normalization | Batch normalization (generator), instance normalization (discriminator) |
| Activation function | ReLU (generator), Leaky ReLU (discriminator) |

## Experimental settings

We use Python to run and train the learning network. All experiments are completed in the hardware environment of gtx2080 (video memory 6 g). Table 2 shows the training parameters for the Pix2Pix model.

## Results

Figure 4 demonstrates the weight diagram. The horizontal line denotes the emotional weights, and the vertical line indicates the corresponding samples. In the process of generating the initial value of the green line, the eight points on the plot represent the best value of the initial value of the plot. With the increase in the number of iterations, the computer-generated data is approaching the real data, and the layout pattern is gradually becoming mature.

The pix2pix model used in this study adopts the method of random initialization, that is, random selection rather than human designation. Once correctly selected, the model can quickly converge and generate ideal results. On the contrary, it will lead to the dilemma of model network optimization. From the relevant content of Fig. 4, we know that the three essential parameters of the pix2pix model are strictly uncertain. In the process of the experiment, we cannot accurately know how much the learning rate is the best, how to control the number of iterations that can achieve the goal quickly and accurately, and

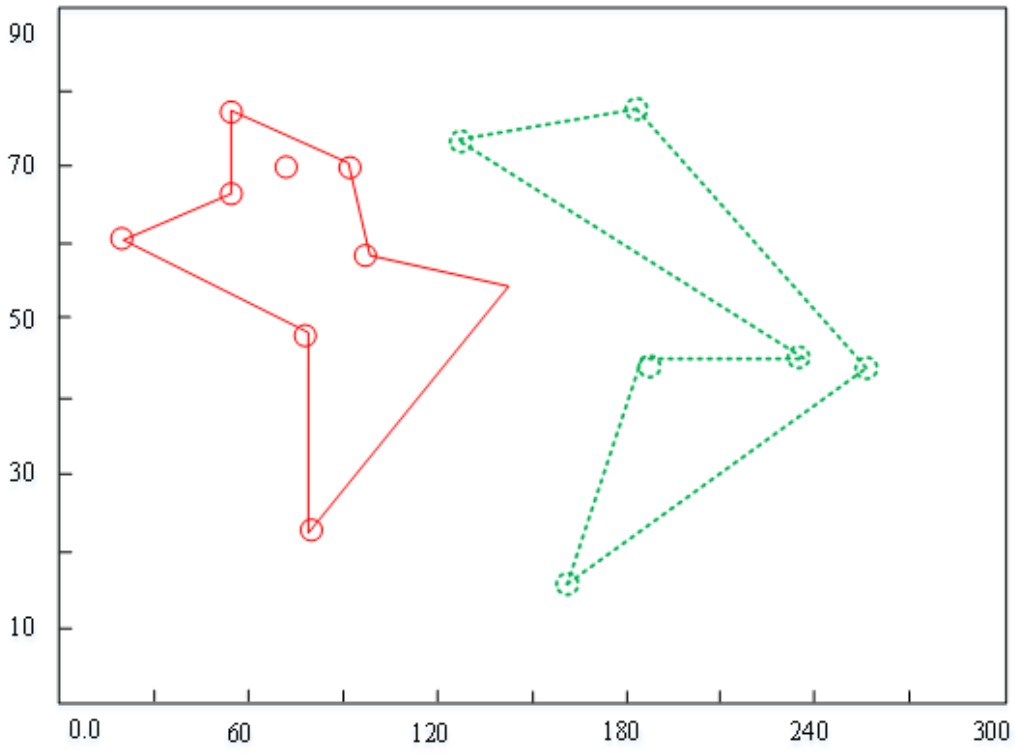

**Figure 4  Weight diagram.**

how to give the initial weight value to optimize the results. These parameters are only one of many variables which require a lot of experience and randomness. The experimental results show that the absolute average error of URDDL is increased by 30.14%, the average absolute percentage is reduced by 11.34%, and the root mean square error is increased by 13%. Similarly, when the type or quantity of training data changes, the changing trend of the loss function is shown in Fig. 5. The horizontal line denotes the landscape flows. The vertical line indicates the change trends.

Regarding the two parameters—learning rate and number of iterations—the minimum value of the loss function may vary with an increase in the number of training samples, and the optimal learning rate can also change. If training continues with a fixed learning rate of 0.0002, one potential issue could be that this rate is too low, preventing the global loss function from reaching its minimum by the end of the iteration. In such cases, potential solutions include either adjusting the learning rate while keeping the number of iterations fixed or increasing the number of iterations while maintaining the current learning rate.

Alternatively, if the learning rate is too high, it may cause the optimization process to skip over the global minimum, resulting in a high loss function or even model instability. To address this, one might reduce the learning rate while fixing the number of iterations. Given that the optimal learning rate is not easily determined and that increasing the number of iterations extends the experimental time, we ultimately chose to optimize the experiment by fixing the number of iterations and adjusting the learning rate accordingly. In conclusion,

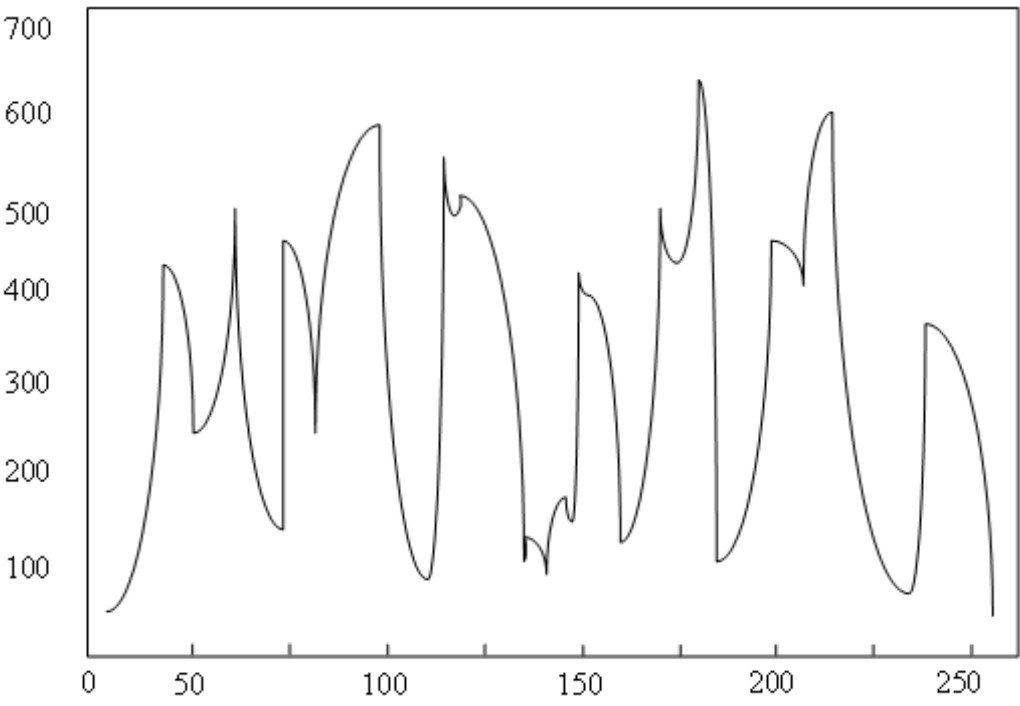

**Figure 5** Change trend of loss function.

the absolute average error of the URDDL model is 91.31%, and the maximum value is 96.87%. The fitting degree is very suitable for evaluating the emotional prediction of urban landscapes.

## Discussion

The implementation of the URDDL model holds considerable promise for architecture and landscape design, as evidenced by its performance metrics and adaptability. The model's capacity to achieve an absolute average error of 91.31%, with a maximum error of 96.87%, underscores its precision in predicting emotional responses to urban landscapes. This precision is of paramount importance for architects and landscape designers who seek to create environments that elicit specific emotional reactions. Accurate prediction of how various landscape elements affect emotional states enables designers to make more informed decisions, thereby enhancing user satisfaction and engagement with their designs.

Another notable feature of URDDL is its adaptability to different types and quantities of training data, as illustrated in Fig. 5. The model's responsiveness to variations in data types and quantities demonstrates its flexibility and versatility, allowing it to be applied across diverse urban contexts and landscape designs. This adaptability makes URDDL a valuable tool for designers working on a wide range of projects, enabling them to tailor their approaches based on the specific characteristics of the landscape and user preferences.

The challenges associated with optimizing the URDDL model, particularly concerning learning rates and iteration numbers, highlight the complexities involved in model tuning. The decision to fix the number of iterations while adjusting the learning rate addresses the

trade-offs between computational efficiency and model accuracy. For architectural and landscape design applications, this optimization strategy ensures that URDDL provides robust and reliable predictions, allowing designers to experiment with various design parameters and refine their approaches based on the insights provided by the model. Furthermore, URDDL's ability to predict emotional responses can be seamlessly integrated into the design process, facilitating continuous evaluation and improvement of urban landscapes. By employing the model to test different design scenarios, architects and planners can gain a deeper understanding of how modifications in landscape features impact public perception and emotional well-being. This iterative approach ensures that design decisions are informed by data-driven insights, aligning them more closely with user expectations.

## CONCLUSION

In conclusion, the URDDL model demonstrates a significant advancement in the field of architecture and landscape design by leveraging deep learning techniques to predict emotional responses to urban landscapes. The model's high precision, as indicated by its low absolute average error, ensures that it can provide reliable insights into how different landscape elements influence emotional states. This capability allows architects and landscape designers to make data-driven decisions that enhance user satisfaction and engagement. The adaptability of URDDL to various types and quantities of training data further underscores its versatility, making it applicable across a broad range of urban contexts and design projects. The model's ability to optimize learning rates and iteration numbers, despite inherent challenges, provides a robust framework for refining design parameters and improving design outcomes. The URDDL model represents a significant advancement in utilizing data-driven methods for emotional prediction in urban landscape design, contributing to the development of more responsive and engaging urban environments. The successful implementation of URDDL also sets the stage for future research in the field of emotional analytics and its application to architecture and landscape design. Future studies could focus on refining the model's parameters and expanding its capabilities to encompass more complex emotional and environmental factors. Additionally, integrating URDDL with other design tools and technologies could further enhance its utility and effectiveness in practical applications.

## ACKNOWLEDGEMENTS

The authors would like to thank the anonymous reviewers for their valuable comments on this article.

### Funding

The authors received no funding for this work.

## Competing Interests

The authors declare there are no competing interests.

## Author Contributions

- Xiaohu Tang conceived and designed the experiments, performed the computation work, prepared figures and/or tables, and approved the final draft.
- Won-jun Chung performed the experiments, analyzed the data, authored or reviewed drafts of the article, and approved the final draft.

## Data Availability

The data is available at Zenodo: Zhiyu Xu, & Shuqing Zhao. (2023). UBGG-3m: Fine-grained urban blue-green-gray landscape dataset for 36 Chinese cities based on deep learning network (v1.0) [Data set]. Zenodo. https://doi.org/10.5281/zenodo.8352777.

## Supplemental Information

Supplemental information for this article can be found online at http://dx.doi.org/10.7717/peerj-cs.2426#supplemental-information.

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
