# Peer review of "Automated urban landscape design: an AI-driven model for emotion-based layout generation and appraisal"

_PeerJ Computer Science, doi:10.7717/peerj-cs.2426_

## Round 0.1 · original submission · Minor Revisions

Please carefully revise your article in light of the comments of the experts and resubmit, please also take into consideration the following comments of mine as well

AE Comments:

The terms "emotional tendency" and "appraisal of an urban landscape" need clearer definitions

The paper presents specific accuracy metrics (e.g., 91.31% average error and 96.87% maximum error), but these figures could be further elaborated
please discuss how user-friendly the interface is?

Discussing the adaptability of the model across various city sizes, climates, and cultural contexts would make the research more broadly applicable

The technical language needs improvements

Reviewer 1 ·

Basic reporting

This research introduces a ground-breaking urban landscape layout model powered by artificial intelligence (AI), which can automatically generate design layouts based on two key dimensions: emotional tendencies and appraisals of urban landscapes, overall the research paper seems good but I strongly suggest to consider the below mentioned points.

1. Title fails to describe research problem under consideration
“Automated urban landscape designing using AI with emotional and appraisal tendencies”
2. Information about the proposed method is omitted from the abstract, authors introduce the problem and provides results.
3. Since the manuscript is interdisciplinary, so it is advised to have a background section for the readers understanding
4. Foreign and domestic deep learning is an ambiguous term~ line 67

Experimental design

5. The last paragraph in the introduction section is not fulfilling the requirements of the manuscript. It should be rewritten in accordance to the research objectives and proposed method
6. Related work has only ten references, and they are not well not connected too.
7. Section 3 URDDL Method is starting from training, perhaps it should describe the components of the modal i.e., generator specification and discriminator specification. The figure 1, is oversimplified it don’t depict anything about generator and discriminator.

Validity of the findings

8. 28 pairs of samples for implementing any flavor GAN are an extreme case of undertraining, it is not going to converge, please check your dataset again
9. Data collection process is not well defined and nothing about training data
10. The lines 211-222 claims are vivid, it should be crux of the entire research work, any deep learning expert needs this information. Unfortunately, either these claims are essentially false or misinterpreted.
11. It is simply unfair to fix learning rate and number of iterations for a landscape modal, lines 270-284 needs special attentions (Optimizing a network which is never converged)

Reviewer 2 ·

Basic reporting

The importance of this research lies in its potential to revolutionize urban landscape design by enhancing efficiency, accuracy, and emotional resonance in the layouts. As demonstrated by experimental results, the model's ability to outperform traditional design methods highlights its applicability in future urban planning and development. However, the incorporation of the following suggestions could further improve the quality of the manuscript
1. Add a discussion section to evaluate the significance of the implementation of GAN for architecture and landscape designing
2. Conclusion should only focus on the implementation of URDDL Modal and its viability in this particular research area
3. UBGG-3m: dataset was used add a section or subsection about the dataset
4. Indicate the specification of the GPU that handles 3-meters resolution images datasets especially the normalization technique before input to the modal
5. The architecture is oversimplified, explain different components in detail in the section 3

Further Comments
6. The abstract is shortened (perhaps journal’s constraint) but at least provide a few sentences about the problem, objective, method, and results.
7. The manuscript is interdisciplinary, to engage the reader authors must provide information about terminologies (landscape designing and deep learning)
8. The first paragraph of the introduction section is dealing with irrelevant material, please be precise and define the research problem
9. Second paragraph of the introduction section is not well connected with the first paragraph
10. The last paragraph is not providing enough information about the following
a. Research objectives
b. Research questions
c. Methods used
d. Organization of the manuscript
11. Related work is not well connected and it only contains ten references, use proper systematic review techniques to find relevant research artifacts.
12. Background section is required for this manuscript
13. Most of the equations have no reference.
14. Overall presentation should be imporved

Experimental design

As above

Validity of the findings

As above

---

## Round 0.2 · Minor Revisions

Dear authors

Thank you for your re-submission, I am pleased to inform you that the experts in the field technically accept your manuscript. Although, the manuscript is scientifically sound and acceptable for publication, but the paper should also be good in technical writing, therefore, we request you to please revise the technical language of the article and re-submit. Following are some suggestions regarding corrections, but please take a last look at the overall language:

Line 54 "With the interdisciplinary " can be With interdisciplinary
Line 55 "In the end of 20th century" can be At the end of the 20th century
Line 62 mode through combing the progress of western can be mode by combing the progress of the western
Line 69 disunity of wetland ecosystem can be disunity of wetland ecosystems

Reviewer 1 ·

Basic reporting

The review comments are well addressed

Experimental design

The experimental design is now well structured

Validity of the findings

The article as a whole looks good and can be accepted

Reviewer 2 ·

Basic reporting

The authors have done a great job to revise the manuscript.

Experimental design

The authors have done a great job to revise the manuscript.

Validity of the findings

The authors have done a great job to revise the manuscript.

---

## Round 0.3 · accepted · Accept

Dear authors,

Based on your revised submission. I have evaluated it and am happy to let you know that we are now recommending your paper for publication. Thank you for your fine contribution.